# Antigenic variation is caused by long plasmid segment conversion in a hard tick-borne relapsing fever *Borrelia miyamotoi*

Tomohi Takeuchi[1], Yasuhiro Gotoh[2], Tetsuya Hayashi[3], Hiroki Kawabata[4], Ai Takano [1]*

1 Department of Veterinary Medicine, Joint Graduate School of Veterinary Medicine, Yamaguchi University, Yamaguchi, Japan, 2 Advanced Genomics Center, National Institute of Genetics, Mishima, Shizuoka, Japan, 3 Department of Bacteriology, Faculty of Medical Sciences, Kyushu University, Fukuoka, Japan, 4 Department of Bacteriology-I, National Institute of Infectious Diseases, Shinjyuku-ku, Tokyo, Japan

* a-takano@yamaguchi-u.ac.jp

## Abstract

*Borrelia miyamotoi* is a hard tick-borne spirochete genetically related to relapsing fever *Borrelia* and the etiological agent of an emerging infectious disease in humans. Like relapsing fever *Borrelia*, *B. miyamotoi* carries clusters of gene cassettes encoding variable major proteins (Vmps) on multiple linear plasmids and shows antigenic variation in mammalian hosts by switching the expression *vmp* gene cassette. However, it remains unknown how the switch occurs in *B. miyamotoi*. Here we determined the whole genome sequences of Japanese *B. miyamotoi* strains to identify the repertoire and arrangement of *vmp* gene cassettes on five linear plasmids, and based on this information, analyzed *B. miyamotoi* clones reisolated from experimentally infected mice. Our analyses revealed that the switch occurred by replacing the expression cassette and its downstream silent cassettes with the long segment from archival plasmid. As the result of this long segment conversion, the first cassette became the expression cassette. Notably, this phenomenon was not due to single gene conversion but the replacement of a long (up to 16 kb or more) plasmid segment. We also show that while bacterial elimination depended on the presence of specific antibodies, the segment conversion was detected at five days post-infection, earlier than antibody production in mice, and even in severe combined immunodeficient mice. These results provide novel insights into the mechanisms that *Borrelia* evolved to survive and persist in mammalian hosts.

## Author summary

We demonstrated that in *Borrelia miyamotoi*, a hard tick-borne spirochete, the switch of the expressing Vmp (antigenic variation) is caused by long plasmid segment conversion. This conversion occurred via the replacement of the *vmp*

**Data availability statement:** The data that support the findings of this study are publicly available at PRJDB10961: https://www.ncbi.nlm.nih.gov/bioproject/?term=PRJDB10961 PRJDB20525: https://www.ncbi.nlm.nih.gov/bioproject/?term=PRJDB20525.

**Funding:** This study was supported by the Japan Agency for Medical Research and Development (AMED) under grant number 23fk0108614h (AT) and JST SPRING under grant Number JPMJSP2111 (TT). The funders had no role in study design, data collection and analysis, decision to publish, or preparation of the manuscript.

**Competing interests:** The authors have declared that no competing interests exist.

expression locus on a linear plasmid with a segment on another linear plasmid where a cluster of silent *vmp* gene cassette is encoded, contrasting to the mechanism in soft tick-borne spirochetes where antigenic variation occurs by the conversion of a single *vmp* gene. We also show that this conversion occurs at an early stage of infection, even at five days post-inoculation, in mice. Thus, this study reveals a unique bacterial system to evade mammalian immunity.

## Introduction

The genus *Borrelia* are arthropod-borne spirochetes, and some *Borrelia* species cause Lyme disease (LD) or relapsing fever (RF) in humans. *Borrelia* can be classified into three phylogenetic groups: LD borreliae, RF borreliae, and reptile-associated borreliae [1,2]. In addition to the phylogenetic differences, the vectors of each group are different: LD *Borrelia* are transmitted by genus *Ixodes* ticks, reptile-associated borreliae are detected from genera *Amblyomma* or *Hyalomma* ticks, and RF *Borrelia* are mainly transmitted by soft-bodied ticks of the genera *Ornithodoros* or *Argas*, with an exception of *B. recurrentis*, which is transmitted by lice. Furthermore, a novel group of RF *Borrelia* was recently reported as a monophyletic hard tick-borne RF borreliae lineage [3]. *Borrelia miyamotoi*, the representative species of this group, was originally isolated from hard-bodied ticks, *Ixodes persulcatus*, and wild rodents, *Apodemus argenteus*, on Hokkaido Island, Japan [4]. Subsequently, human cases of *B. miyamotoi* infection were reported in Russia in 2011 as the causative agent of emerging relapsing fever, followed by cases in Asia including Japan, Europe, and the United States [5–8]. Because phylogenetic analysis indicated that hard tick-borne RF *Borrelia* are ancestral species of RF *Borrelia*, understanding their pathogenetic mechanisms is an important step to understand the evolution of the pathogenicity of RF *Borrelia*.

Soft tick-borne RF *Borrelia* (STRF *Borrelia*) infections in humans cause a series of febrile episodes between recovery periods by antigenic variation of spirochetes. This complex immune evasion strategy creates new serotype populations that evade the humoral immunity induced by the initially infected *Borrelia* [9,10]. It is known that several viruses, bacteria, fungi, and parasites that evade immune clearance by antigen switching via gene conversion, point mutations, reassortment and epigenetic changes [11–16]. Host antibodies were directed against a surface antigenic protein that determines the serotype of the infecting population [17–19]. The immunogenic surface proteins of RF *Borrelia* are designated as variable major proteins (Vmps). RF *Borrelia* carries a single linear chromosome, multiple circular plasmids, and multiple linear plasmids on which *vmp* gene cassettes are distributed [20,21]. In *B. hermsii*, a STRF *Borrelia*, a single expression locus where an expression *vmp* cassette with an active promoter is located at the end of a linear plasmid, lp28–1, and many silent *vmp* cassettes are preserved in other linear plasmids as transcriptionally inactive pseudogenes [13,22–26]. As *vmp*-disrupted mutants of *B. hermsii* showed a reduced bacterial load in mouse blood and induced no recurrent bacteremia, Vmp production prolongs bacterial survival in blood and contributes to the infectivity of *B. hermsii* in experimentally infected mice [17,27].

Vmps can be classified into two major groups by molecular size: approximately 25 kDa proteins termed variable small proteins (Vsps) and approximately 37 kDa proteins termed variable large proteins (Vlps) [26]. Antigenic variation occurs via a gene conversion event such that a silent *vmp* cassette is translocated downstream of the promoter at the expression locus, resulting in a new serotype population [9,22]. For example, inoculation of *vlp7-* or *vlp17-*expressing *B. hermsii* into mice results in the conversion of the major serotype to other serotypes, such as *vlp2*, *vlp24*, and *vlp13* [28]. The mechanism of *vmp* gene conversion has been studied in several STRF *Borrelia*, particularly in *B. hermsii* [13]. The conversion occurs between upstream homologous sites (UHS) and downstream homologous sites (DHS) conserved in the expression cassette and silent *vmp* cassettes [17,29]. Because the replacement of the expression *vmp* cassette and a single silent *vmp* cassette on a different plasmid occurs between the UHS and DHS, the length of the plasmid where the expression locus is located does not change drastically when gene conversion occurs.

The *vmp* genes in *B. miyamotoi* with homology to *B. hermsii vmp* genes were first reported by Hamase et al. [30] and confirmed by DNA sequencing in a study by Barbour [21]. The conversion of Vmp protein in *B. miyamotoi* was first demonstrated in vitro by Wagemakers using the North American strain LB-2001. Most bacteria expressing Vsp1 are killed by specific anti-Vsp1 antibody-mediated selection, resulting in a new bacterial population expressing VlpC2 [31]. It has also been demonstrated that *B. miyamotoi* LB-2001 was killed by the anti-*B. miyamotoi* antiserum, resulting in a new serotype population due to the switch of Vmp protein expressed [32]. In addition, the antigenic variation of *B. miyamotoi* has been shown in experimentally infected mice [33]. However, the actual mechanism underlying antigenic variation in *B. miyamotoi* is unclear owing to difficulties to re-isolate *B. miyamotoi* from infected mouse tissues or whole blood. Herein, to further characterize the antigenic variation of *B. miyamotoi* in mammalian body, we injected *B. miyamotoi* into mice and analyzed re-isolated strains to elucidate how the switching of the expression *vmp* cassette occurs.

## Results

### Genome sequencing of the two cloned Japanese *B. miyamotoi* strains

To determine the repertoires and genetic organizations of *vmp* gene cassettes, we sequenced two cloned Japanese strains, MYK1 clone G3 and M1-2Br clone H4 using the PacBio RS II system and Illumina MiSeq or HiSeq sequencer (S1 Table) [3]. MYK1 was isolated from *I. pavlovskyi* in 2012 and cloned by limiting dilution. None of the eight clones of MYK1, including clone G3, caused bacteremia when 2 x 10⁵ cells were inoculated intraperitoneally into C57BL/6J (referred to as B6) mice. M1-2Br was isolated from the brain tissue of a C57BL/6J mouse inoculated intraperitoneally with 2 x 10⁵ cells of a low-passage MYK1 (non-clonal) strain (see Method for the isolation procedure) and cloned by limiting dilution. All of the five established clones caused bacteremia in B6 mice.

By this sequencing, we obtained almost complete genome sequences of the two clones, MYK1 clone G3 (AP024375-AP024390) and M1-2Br clone H4 (AP024356-AP024370). In addition to linear chromosomes in each clone, we obtained 15 plasmids, including 12 linear and three circular plasmids, from MYK1 clone G3 and 14 plasmids, including 12 linear and two circular plasmids, from M1-2Br clone H4 (S2 Table). As for the chromosomes and linear plasmids, telomere-to-telomere sequences were obtained for both clones, except for two plasmids of M1-2Br clone H4 (lp3 and lp10.1), for which left end sequences were not obtained. The genome sequences of the two clones were highly conserved (S1 Fig), except that M1-2Br clone H4 lacked a circular plasmid (cp8.2) and a 4,593-bp relative deletion was present in a linear plasmid (lp4). Of the plasmids identified, five linear plasmids (lp4, lp5, lp6, lp10.1, and lp11) contained various *vmp* gene cassettes. The *vmp* expression locus with the promoter sequence documented by Barbour in LB-2001 [21] was present on lp4, where the relative deletion was found.

### The repertoire and genetic organization of *vmp* gene cassettes in M1-2Br clone H4

Through *in silico* analysis, we identified 44 *vmp* gene cassettes on the five linear plasmids in M1-2Br clone H4, which were classified into 22 *vlp*, 11 *vsp*, and 11 untypeable genes (most were presumably pseudogenes) (Fig 1). In this

article, *vmp* gene cassettes were denoted according to their locus tags in the genome of M1-2Br clone H4 (for example, BmHJ_k00016 is referred to as Jk16). Of the 22 *vlp* genes, while 6 and 14 belonged to the *vlp*-C and *vlp*-D subfamilies, respectively, only one gene was found for both the *vlp*-A and *vlp*-B subfamilies, showing a contrast to the STRF *Borrelia* (S2 Fig). The *vmp* gene cassettes existed as clusters of various numbers (two to 12 cassettes) on each plasmid, including lp4, where the expression cassette (Jc14) with a promoter sequence was clustered with four silent cassettes located in its downstream region (Fig 1). Notably, all cassettes in each cluster were organized in the same orientation, contrasting to the organization of *vmp* cassettes in STRF *Borrelia* [13].

Comparison of the expression loci of MYK1 clone G3 and M1-2Br clone H4, where a 4,593-bp relative deletion existed, revealed that in MYK1 clone G3, the expression cassette and a following silent cassette (Jc14 and Jc12, respectively) were missing and the third cassette (Jc7) in the cassette cluster was placed just downstream of the promoter (S1 Fig). Southern blot analysis using a promoter sequence-specific probe confirmed that the expression locus is present on a single plasmid (referred to as the *vmp*-expression plasmid) in both clones (S3 Fig).

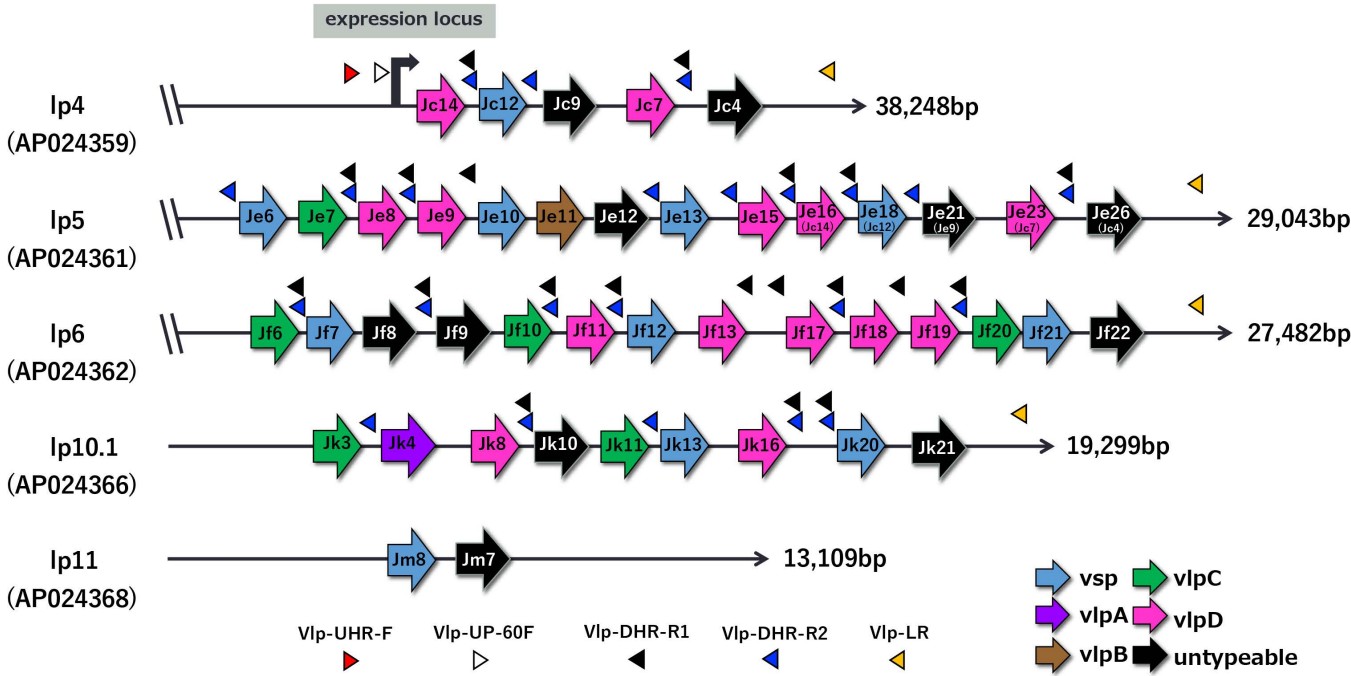

**Fig 1. The genetic organization of *vmp* expression locus and silent *vmp* cassette clusters on five linear plasmids in *B. miyamotoi* M1-2Br H4.** The locations and genetic organizations of expression *vmp* cassette and silent cassettes on five linear plasmid of *B. miyamotoi* M1-2Br strain are schematically shown. Vmp proteins are classified into two major families; variable small proteins (Vsps) and variable large proteins (Vlps). The Vlps are further classified into 4 groups, *vlp*-A, *vlp*-B, *vlp*-C and *vlp*-D, which are indicated by purple, brown, green, and pink color respectively. Arrowheads indicate the primer sites. The Vmp-UHR-F and Vmp-UP-60F primers designed at the upstream region of promoter are specific to lp4. The Vmp-DHR-R1 and Vmp-DHR-R2 primers were both designed in the sequence conserved in multiple intergenic regions of *vmp* cassettes. As there was some variation in the sequence in intergenic regions, two primers were designed. Therefore, the expression *vmp* cassette can be amplified using Vmp-UHR-F as the forward and either Vmp-DHR-R1 or Vmp-DHR-R2 as the reverse primer. Vmp-LR was designed in the short-conserved sequence identified at the right end of the lp4, lp5, lp6 and lp10.1 plasmids. Vmp-UP-60F was used as a sequence primer. Parentheses under the plasmid name indicate the accession numbers of their sequences. Plasmid sizes are shown on the right of each plasmid.

PLOS Pathogens

## Sequence comparison of *vmp* cassette-bearing plasmids of M1-2Br clone H4

Sequence comparison of the five *vmp* cassette-bearing linear plasmids of M1-2Br clone H4 revealed that the right end sequences of three plasmids (lp4, lp5, and lp6) were highly conserved (S4 Fig). In particular, the 11,574-bp sequences of lp4 starting from the expression *vmp* cassette (Jc14) and ending at the telomere was 100% identical to the right end sequence of lp5 (from Je16 to the telomere). Therefore, the expression cassette (Jc14) and four downstream silent cassettes (Jc12, Jc9, Jc7, and Jc4) on lp4 were identical to Je16, Je18, Je21, Je23, and Je26 on lp5, respectively. The 4,027-bp right end sequence of lp6 was also 100% identical to the corresponding sequences of lp4 and lp5. In addition, parts of the 4,027-bp sequence were also conserved in lp10.1, but with some sequence diversity. The remaining *vmp* cassette-bearing linear plasmid, lp11, caried only two cassettes and its right end sequence was different from those of the other four plasmids, although a 1,071-bp sequence upstream of the two cassettes was 100% identical to the sequence upstream of the cassette cluster of lp10.1.

## Search for *B. miyamotoi* strains suitable for the analysis of strains reisolated from mice

Initially, we planned to reisolate strains (subclones) from mice infected by M1-2Br clones and analyze their expression *vmp* cassettes. Therefore, we first determined the expression *vmp* cassette in the remaining four clones of the M1-2Br by sequencing the PCR amplicons obtained with a forward primer (Vmp-UHR-F; designed on the upstream of the promoter region; thus specific to lp4) paired with reverse primers (Vmp-DHR-R1 or Vmp-DHR-R2, both were designed on multiple intergenic regions of silent cassettes) (Fig 1). Of the four M1-2Br clones, three carried the same expression cassette (Jc14) as that of clone H4, but clone F5 carried Jk16. We then inoculated these clones to B6 mice to obtain re-isolated strains. However, although bacteremia was observed for all of the five M1-2Br clones, we could not re-isolate strains from any of the five clones. Therefore, to search for alternative *B. miyamotoi* strains that can be used for this purpose, we analyzed other Japanese *B. miyamotoi* strains stored in our laboratory for their expression *vmp* gene cassettes by the above-mentioned strategy and abilities to cause bacteremia by inoculation to B6 mice. By this search, we found three strains (non-clonal) that expressed Jk16 or Je16 (=Jc14) and caused bacteremia. After cloning by limiting dilution and determining the expression *vmp* cassettes of each clone, we selected MYK2 clone A9 (referred to as "MYK2 A9 strain", expression cassette; Jk16) and MYK4 clone C8 (referred to as "MYK4 C8 strain", expression cassette; Je16/Jc14). Reverse transcription (RT)-PCR analysis confirmed that these clones expressed Jk16 and Je16/Jc14, respectively. As we confirmed, in preliminary experiments, that re-isolation of these strains from infected mice was possible, we performed whole-genome sequencing of both strains and confirmed that these strains shared the same repertoire and genetic organization of archival *vmp* gene cassettes on lp5, lp6, lp10.4, and lp11 with M1-2Br H4 (S5 Fig). Therefore, MYK2 A9 and MYK4 C8 were used for subsequent analyses.

## Copy numbers of two cloned *B. miyamotoi* strains, MYK2 A9 and MYK4 C8, in mice

MYK2 A9 and MYK4 C8 were injected into B6, BALB/cJ, BALB/c-nude (nude), and CB17SCID (severe combined immunodeficient (SCID)) mice, respectively, and whole blood samples were collected to calculate *B. miyamotoi* burden using real-time quantitative PCR (qPCR) (S6 Fig). In B6 mice, bacteremia was observed in all mice (4/4) at 5 days post-inoculation (dpi) and in two (2/4) mice at 10 dpi for both strains (Fig 2). Thereafter, bacteria were not detected in all mice used in this experiment (15, 20, and 28 dpi). In BALB/cJ mice, bacteremia was observed in all mice (4/4) at 5 dpi for both strains. At 10 dpi, it was observed in three (3/4) MYK2 A9-infected mice and all MYK4 C8-infected mice (4/4). At 15 dpi, while bacteria were detected in none (0/4) of the MYK2 A9-infected mice, it was observed in three MYK4 C8-infected mice (3/4). Thereafter, bacteria was not detected in all mice (20 and 30 dpi). In most nude mice, bacteremia was developed until 20 dpi. Thereafter, the bacterial count was below the detection limit in all the mice used in this experiment (30 dpi). In contrast to these mice, persistent bacteremia was observed in all SCID mice up to 31 dpi for both strains. Moreover, among the

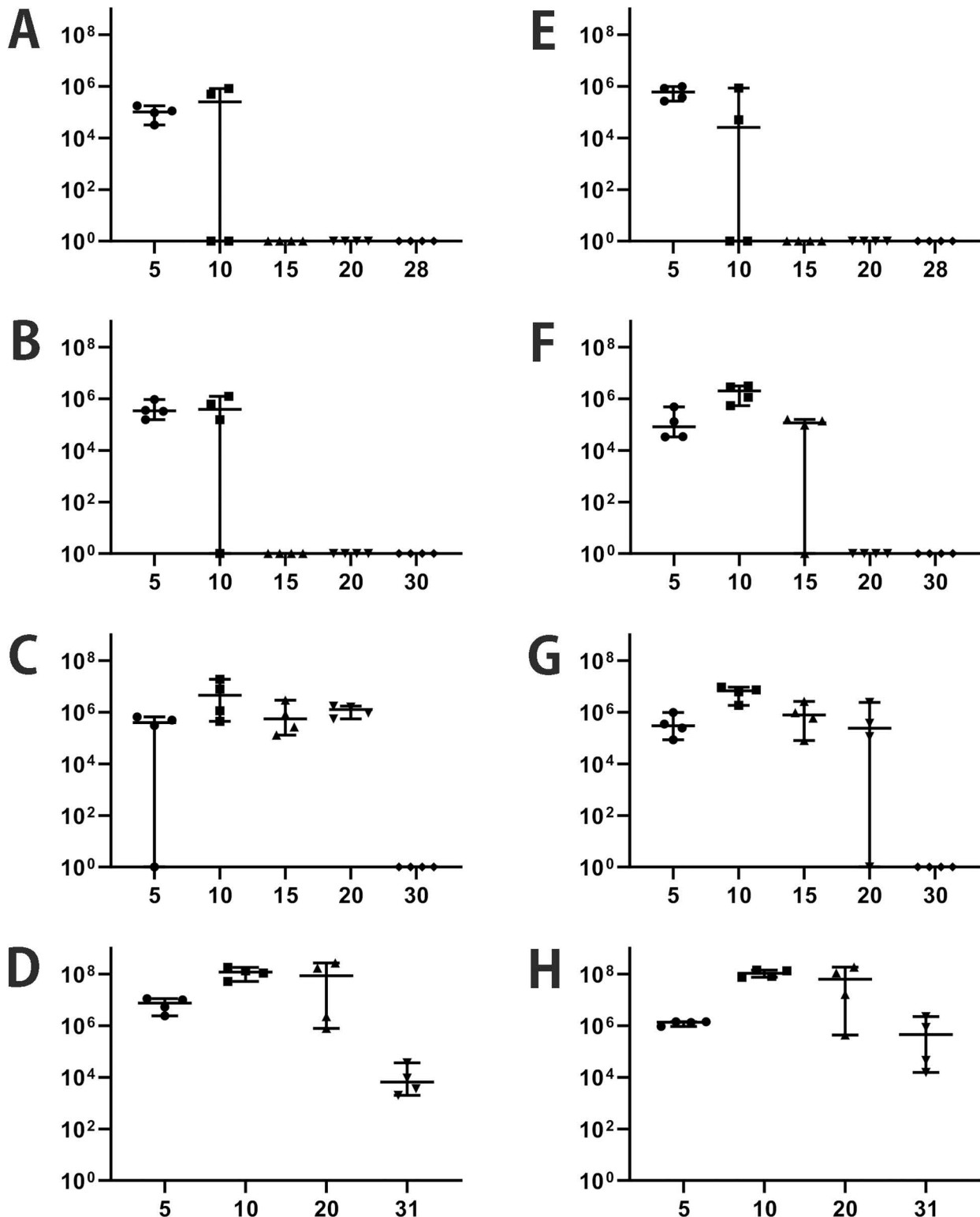

**Fig 2. Bacteremia levels in *B. miyamotoi*-injected mice.** The bacteremia level in C57BL/6J (panels A and E), BALB/cJ (B and F), BALB/c-nude (C and G), and CB17SCID (D and H) mice are shown. A-D are the data of MYK2 A9-injected mice and E-H are the data of MYK4 C8-infected mice. The horizontal axis indicates the days post-inoculation, and the vertical axis indicates the copy number of *B. miyamotoi* 16S rDNA per ml of mouse whole blood. As *B. miyamotoi* contains only one *rrn* operon, the copy number of 16S rDNA roughly represents the number of *B. miyamotoi* bacterial cells.

mice analyzed, SCID mice at 10 dpi showed the highest bacterial count (approximately $10^8$ bacteria/mL) although bacterial cell counts gradually decreased after 20 dpi (Fig 2). None of the mice exhibited any clinical signs in this study.

### Expression *vmp* cassette changes in *B. miyamotoi* reisolated from experimentally inoculated mice

To analyze the change in the expression *vmp* cassette in mammalian body, we tried to re-isolate MYK2 A9 and MYK4 C8 from liver or whole blood samples of infected mice. We performed two independent infection experiments using B6 mice and obtained 12 and four re-isolates at 5 and 10 dpi, respectively (S3 Table). Of the 16 re-isolates, four (B6M-1w, B6M-4L, B6M-9L, and B6M-11w) obtained at 5 dpi and two (B6M10-22w and B6M10-28L) obtained at 10 dpi were cloned again by limiting dilution. Of these six re-isolates, three (B6M-1w, B6M-4L, and B6M10-22w) were isolated from MYK2 A9-inoculated mice, and the remaining three (B6M-9L, B6M-11w, and B6M10-28L) were isolated from MYK4 C8-inoculated mice. The expression *vmp* cassette of these re-isolates had become heterogenous. Therefore, these re-isolates were cloned by limiting dilution, followed by analysis of expression *vmp* cassette (see S6 Fig left panel and S3 Table). Of the clones derived from three re-isolates of MYK2 A9 (originally expressing Jk16), Jk8 (14/23) or Jf17 (9/23) was found at the expression site in the clones of B6M-1w; Jf11 (12/13) or Jk8 (1/13) was found in the clones of B6M-4L; only Jf6 (16/16) was found in the clones of B6M10-22w. Of the three re-isolates of MYK4 C8 (originally expressing Je16/Jc14), Jk8 (9/19), Jk4 (5/19), Je23 (3/19), Jf11 (1/19), or Je15 (1/19) was found in the clones of B6M-9L; Jf18 (15/16) or Jf17 (1/16) was found in the clones of B6M-11w; Jf11 (15/19) or Je16 (4/19) was found the clones of B6M10-28L (S3 Table). These results indicated drastic and highly variable changes in the expression cassette among the re-isolates.

### Direct detection of expression *vmp* cassettes in the DNA preparations obtained from whole blood or liver samples of *B. miyamotoi*-infected mice

To eliminate bias due to cultivation, we further examined the changes and variations in expression *vmp* cassette generated in infected mice by PCR amplification and sequencing of expression cassettes present in the DNA purified from mouse samples. In this analysis, we first inoculated MYK2 A9 and MYK4 C8 into B6 mice and collected whole blood or liver sample at 5 dpi or 5 and 10 dpi. After PCR amplification of expression cassette as described above, PCR products were subjected to TA cloning and multiple clones (4–21 clones from each sample) were sequenced to estimate the proportions of each *vmp* cassette carried at the expression site (S7 and S8 Figs). In both MYK2 A9-inoculated and MYK4 C8-inoculated B6 mice, drastic changes in expression cassette were observed from 5 dpi and clones encoding the original expression cassettes (Jk14 or Je16/Jc14) were not detected (Fig 3A and 3E). Although this analysis detected the *vmp* cassettes detected in the analysis of re-isolates (S3 Table), the PCR/TA cloning method detected more cassettes, thus is apparently more sensitive than the re-isolation strategy.

We performed similar analyses using BALB/cJ, nude, and SCID mice (Fig 3B-3D and 3F–3H; see S7 and S8 Figs for the frequencies of expression cassettes detected in each sample). In BALB/cJ mice, similar to B6 mice, drastic changes in expression cassette were observed from 5 dpi and clones encoding the original expression cassettes were not detected. In nude mice, the change in expression cassette was observed from 5 dpi in all MYK4 C8-inoculated mice (3/3) and one MYK2 A9-inoculated mouse (1/3). In SCID mice, although no expression cassette changes were detected in the MYK2 A9-inoculated mice (2/2) and one of the MYK4 C8-inoculated mouse (1/2) at 5 dpi, an altered expression cassette was detected in the other MYK4 C8-inoculated mouse. At 10 dpi, while the original cassette was still detected in one of the two MYK2 A9-inoculated mouse (only one clone out of the 12 clones analyzed), various cassettes were detected both in MYK2 A9-inoculated and MYK4 C8-inoculated mice. The original expression cassettes were no more detected at 20 and 31 dpi in all SCID mice.

### Mechanism underlying the antigenic variation in *B. miyamotoi*

To understand the mechanism underlying the antigenic variation (changes in the expression *vmp* cassette) in *B. miyamotoi*, we first searched the M1-2Br H4 genome for the sequences corresponding to the UHS and DHS which are

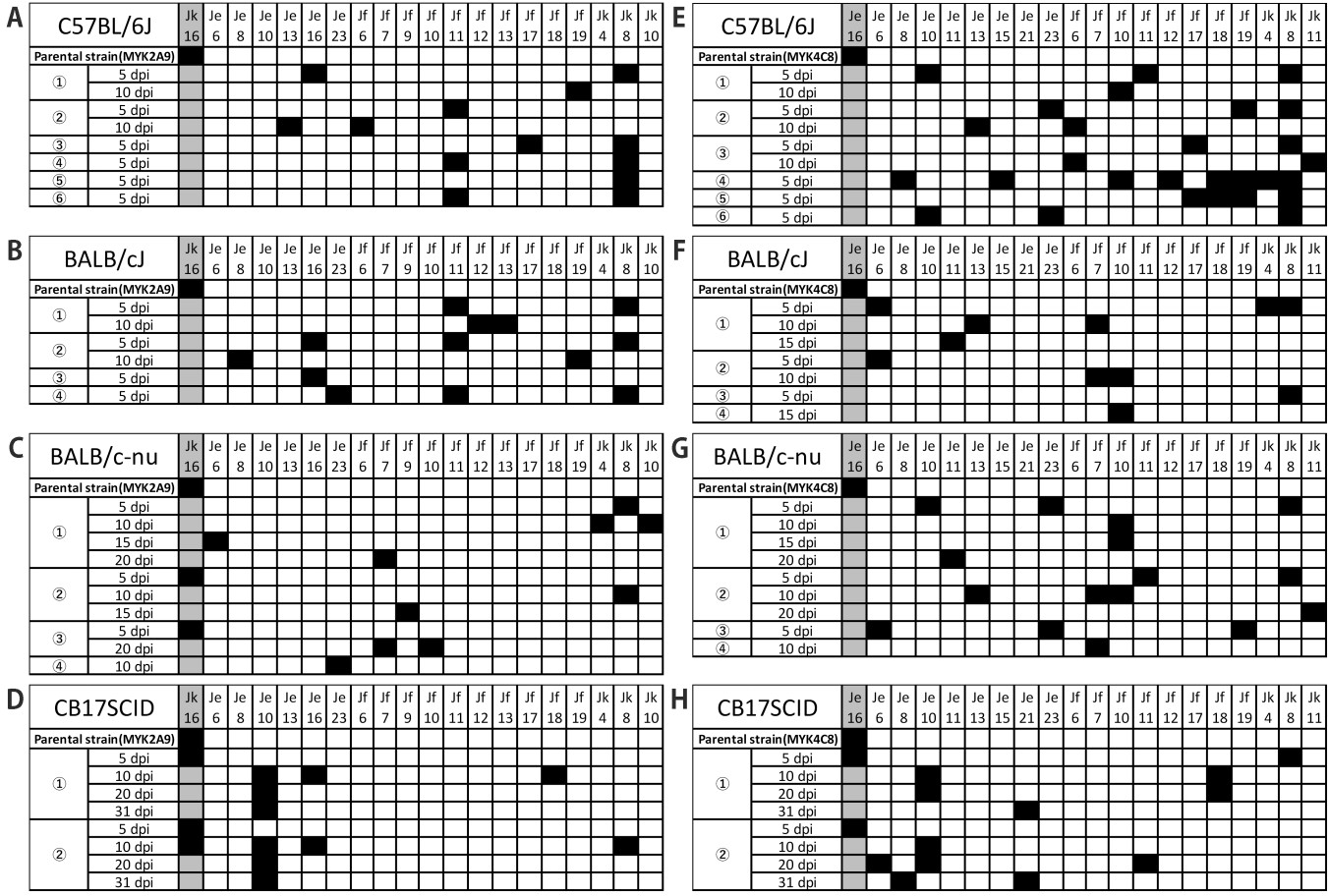

**Fig 3. Changes in expression *vmp* cassette in *B. miyamotoi* MYK2 A9 (from A to D) and MYK4 C8 (from E to H) after infection to C57BL/6J, BALB/cJ, BALB/c-nude, or CB17SCID mice.** The *vmp* cassettes detected in mouse samples by PCR-based TA cloning are indicated. Circled numbers indicate individual mouse numbers. The data of C57BL/6J is shown in A and E; of BALB/cJ in B and F; of BALB/c-nude in C and G; of CB17SCID in D and H.

thought to mediate expression cassette changes via gene conversion in STRF *Borrelia* [17,29]. However, while UHS-like sequences were detected (S9 Fig), we did not find DHS-like sequences, raising a possibility that the mechanism in *B. miyamotoi* differs from that in STRF *Borrelia*. Another notable difference between *B. miyamotoi* and STRF *Borrelia* was the location and genetic organization of the expression locus. In STRF *Borrelia*, the expression locus is located at the very end of a linear *vmp* expression plasmid and contains only one expression cassette: thus the length of the *vmp* expression plasmid does not change if the expression cassette is changed via gene conversion. In *B. miyamotoi*, as mentioned already, multiple silent cassettes located downstream of the expression cassette on the *vmp* expression plasmid (lp4). Moreover, the segment identical to the segment starting from the expression cassette and ending at the right end of lp4 was present on one of the four archival silent *vmp* cassette-encoding plasmids (referred to as archival plasmids) in *B. miyamotoi* M1-2Br clone H4. Based on these features, we speculated that a segment of an archival plasmid may be duplicated and translocated to the *vmp* expression plasmid to replace the expression locus (the conversion of long plasmid segment; not the conversion of single gene as observed in STRF *Borrelia*). As a result of this conversion, the first silent *vmp* cassette on the translocated segment is placed just downstream of the promoter, leading to the change in expression cassette.

To verify this hypothesis, we performed long PCR with a forward primer (Vmp-UHR-F; targeting the upstream region of the promoter on lp4) paired with a reverse primer, Vmp-LR, which binds to the right end of lp4, lp5, lp6, and lp10.1 (see Fig 1 for the positions of these primers). As we did not detect the clones expressing Jm7 and Jm8 encoded on lp11 in mouse experiments and the right end of lp11 was structurally different from other *vmp* cassette-encoding plasmids, lp11 was excluded from subsequent analyses. For the long PCR analysis, we selected two reisolates obtained at 5 dpi from MYK2 A9-inoculated B6 mice (B6M-1w C2 and B6M-1w A3) in which the expression cassette changed from Jk16 to Jf17 and Jk8, respectively, and two reisolates obtained at 5 dpi from MYK4 C8-inoculated B6 mice (B6M-9L B2 and B6M-9L A5), in which the expression cassette changed from Je16 to Je23 and Jf11 (S4 Table). The long PCR analysis of these reisolates and their parental strains revealed that the lengths of the segment from the promoter to the right end of lp4 (referred to as "expression locus segment") differed significantly between the reisolates and their parental strains and between the isolates (Fig 4). Partial sequences of the expression locus segments in reisolates, which were obtained by direct sequencing of PCR products, suggested that in the reisolates, the expression locus segment was completely replaced by the segments of one of the three archival plasmids, supporting our hypothesis. Moreover, the observed sizes of long PCR products from each reisolate, which represent the lengths of their expression locus segments, were consistent with the calculated lengths that would be generated if the predicted replacement of expression locus segment occurred (Fig 4). For example, the expression locus segment of reisolate B6M-1w A3 expressing Jk8 was approximately 13 kb in size, which is in a good agreement with the length of the segment from Jk8 to the right end of lp10.1. These results further supported our hypothesis. In addition, the sizes of the *vmp* expression plasmid in the four re-isolates and two additional reisolates (B6M-9L H6 expressing Jk4 and B6M-11w B1 expressing Jf18; both obtained at 5 dpi from MYK4 C8-inoculated mice) and two parental strains, which were estimated by pulsed-field gel electrophoresis (PFGE), were consistent with the sizes calculated based on our hypothesis (Fig 5; note that sizes of other plasmids did not change). In

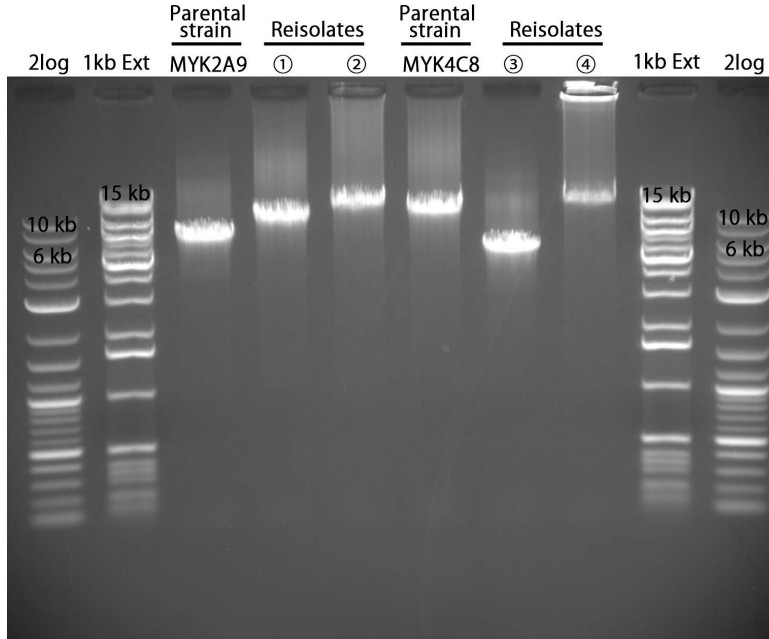

**Fig 4. Changes in size of expression locus segments in parental and re-isolated strains.** Long PCR amplicons from the promoter region to the right end of the *vmp* expression plasmid are shown. Ps; Parent strain. ① and ②; re-isolates obtained at 5 dpi from MYK2 A9-injected mice, B6M-1w C2 and B6M-1w A3, respectively. ③ and ④; re-isolates obtained at 5 dpi from MYK4 C8-injected mice, B6M-9L B2 and B6M-9L A5, respectively. Molecular size markers are denoted on both end of panel in kilobase pairs.

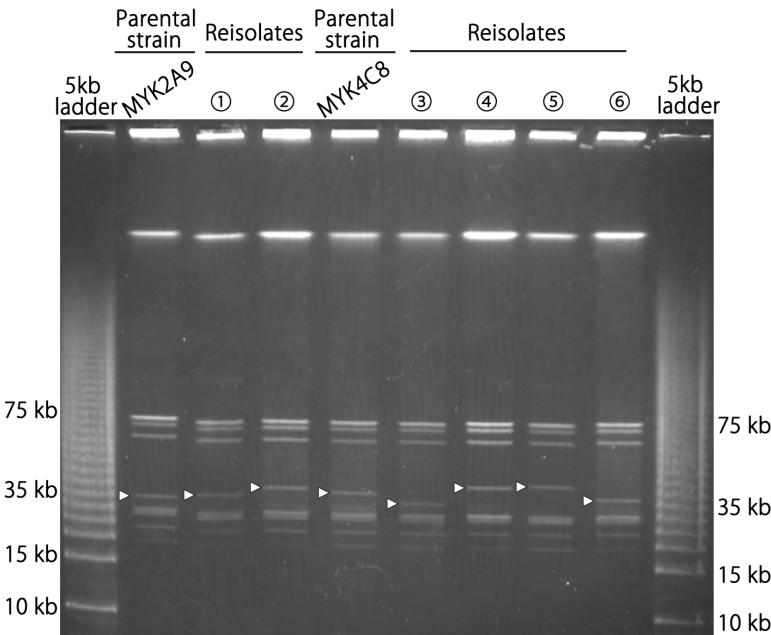

**Fig 5. Whole-genome PFGE of re-isolates and their parental strains.** Whole genome PFGE was performed to compare the sizes of plasmids in parental and reisolates and their parental strains. ① and ②; re-isolates obtained at 5 dpi from MYK2 A9-injected mice, B6M-1w C2 and B6M-1w A3, respectively. ③, ④, ⑤ and ⑥; re-isolates obtained at 5 dpi from MYK4 C8-injected mice, B6M-9L B2, B6M-9L A5, B6M-9L H6 and B6M-11w B1, respectively. Arrowheads indicate the *vmp* expression plasmid. The positions of selected molecular size markers are indicated on both ends of the picture.

the Southern blot analysis of the DNA digested with restriction enzymes using a probe hybridizing to the promoter region in the *vmp* expression site, detected sizes of expression locus segment-containing fragments were also consistent with the sizes calculated based on our hypothesis (S10 Fig).

Finally, to confirm the duplication of the archival plasmid segments that replaced the expression locus segment, we performed whole-genome short-read mapping analysis. In this analysis, the Illumina short reads obtained from the two MYK2 A9-derived reisolates, four MYK4 C8-derived reisolates and two parental strains (the same set of strains used in the PFGE analysis) were mapped to the reference whole-genome sequence of the M1-2Br clone H4. For this analysis, the expression locus segment of the reference sequence was masked. Therefore, if the duplication of an archival plasmid segment that replaced the expression locus segment occurred, two-times more reads would be mapped to the archival plasmid segment than to the remaining regions. As expected, the normalized mapping depth on the segments that were predicted to replace the expression locus segment in each reisolate were roughly two-times higher than that on the remaining regions of each archival plasmid and the corresponding plasmids in parental strains (Fig 6). There were no apparent differences in the mapping pattern for other plasmids between the reisolates and parental strains; as an example, the mapping data for lp1, the longest linear plasmid with no *vmp* gene cassettes, are shown in S11 Fig.

## Discussion

The antigenic variation by gene conversion in *B. hermsii* and *B. turicatae*, members of the STRF *Borrelia* family, have been studied and documented for over 40 years [9,10,34–36]. However, the antigenic variation of hard tick-borne RF *Borrelia* has not been well characterized. In a previous study, the nucleotide sequence analysis of selected plasmids from an American *B. miyamotoi* strain LB-2001 revealed that multiple and diverse *vmp* gene cassettes are encoded on linear plasmids, indicating that *B. miyamotoi* has a genetic composition that allows serotype switching [21]. Later, the full or nearly full set

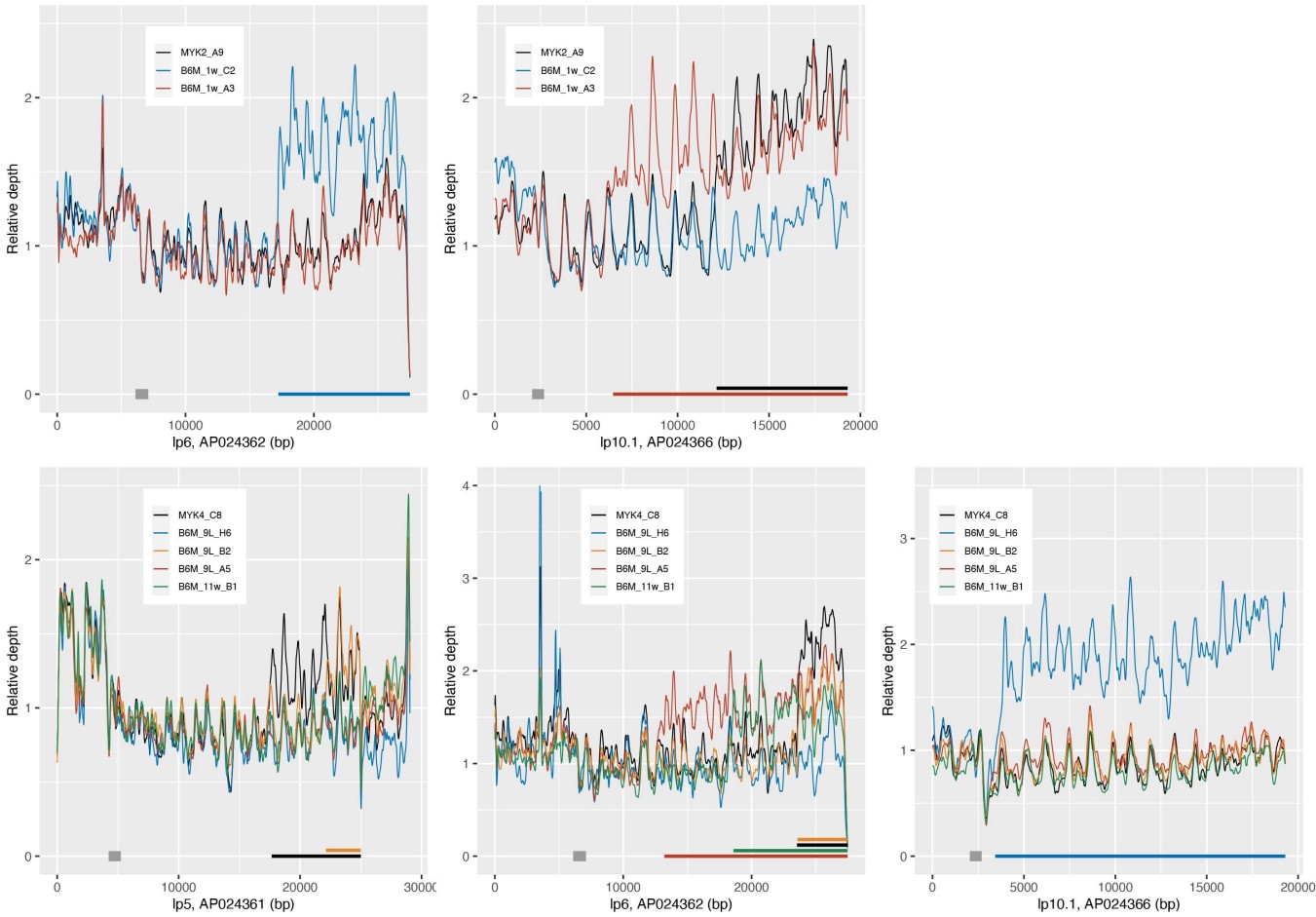

**Fig 6. Mapping of Illumina reads from reisolates and their parental strains to the *B. miyamotoi* M1-2Br H4 genome.** The Illumina reads of reisolates and their parental strains were mapped to the genome of M1-2Br H4. The expression locus segment of the M1-2Br H4 genome sequence (AP024359, 1–11,967 bp) was masked and used as the reference. Mapping depth per base was calculated and normalized using the mean depth of the specified region of each plasmid (AP024361: 4240–5240 bp, AP024362: 6100–7100 bp, AP024366: 2040–2689 bp, indicated by gray rectangles). Mapping depths in 100-bp window sliding every 10 bp were depicted in R ver. 4.2.3 with ggplot2 package ver. 3.5.0. Colored bars indicated on the horizontal axis indicate the regions where a higher number of reads were mapped compared to the other region in each strain. The mapping results of MYK2 A9 and its re-isolates are shown in upper panels. The results of MYK4 C8 and its re-isolates are shown in lower panels.

of *vmp* gene cassettes and their genetic organization of several *B. miyamotoi* strains have been determined by whole genome sequencing in previous studies [37–39] and this study. However, as *B. miyamotoi* is difficult to culture, there is only one report of re-isolation of *B. miyamotoi* from experimentally infected mice [33]. In this report, a tick cell line was used for re-isolation. Because the surface antigens could be changed in ambient environments, re-isolates obtained by the method that does not rely on co-culturing with cells need to be analyzed to understand the antigenic variation occurred *in vivo*. In this study, we successfully established a method to re-isolate *B. miyamotoi* from experimentally infected mice in using Barbour-Stoenner-Kelly (BSK)-M medium (see Materials and Methods for the details). It should be noted that the expression *vmp* cassette was not altered after several passages in BSK-M medium. In addition, we established a PCR-based TA cloning method to detect the expression *vmp* cassettes directly from mice, which reduces bias to detect particular serotypes. Using these methods and based on the genome sequence information, we clearly demonstrated the changes in expression *vmp* gene in infected mice (Fig 3 and S3 Table). Changes occurred as early as 5 dpi, earlier than previously

reported [33], and the bacterial population originally infected was extensively replaced by populations expressing different *vmp* genes in immunocompetent mice. In *B. burgdorferi*, a switch of expressing surface antigen gene *vlsE* has been observed in the early stages of infection [40,41]. This is the first report demonstrating such a phenomenon in *B. miyamotoi*. The detection of organisms expressing the initial *vmp* genotype is prolonged in immunocompromised mice, suggesting that in immunocompetent animals, the expansion of an emerged new serotype population(s) occurs at the early stage of infection via immunological selection. Interestingly, the expression *vmp* cassettes change was not observed in BSK-M medium but was observed in SCID mice, suggesting that factors other than antibodies may play a role in this phenomenon although spontaneous antigenic variation may occur at a very low frequency. The fact that *B. miyamotoi* was not eliminated in SCID mice but was no longer detected in immunocompetent mice after 10 dpi (Fig 2) highlights the importance of antibodies in bacterial elimination. Consistent with this, we also found that IgG levels increased in serum of mice at 10 dpi, as determined by Western blot analysis using whole-cell lysates of *B. miyamotoi* MYK2 clone A9 or MYK4 clone C8 (S12 Fig).

Previously it was shown that when *B. hermsii* was injected into mice, there was some trend in which *vmp* gene is expressed in the newly emerged serotype population [28]. In this study, some trend in the favored *vmp* cassette newly expressed in mice was also observed even though we used two parental strains expressing different *vmp* genes. For example, approximately 34% of the *vmp* cassettes detected at 5 dpi in MYK2 A9- or MYK4 C8-injected B6 mice were Jk8 (S7 and S8 Figs, and S3 Table). A similar bias to Jk8 was also observed in BALB/cJ mice, but it was not obvious in immunodeficient mice.

The most important finding of this study was the demonstration of dramatic changes in expression *vmp* cassette (antigenic variation) at an early stage of infection in *B. miyamotoi*, based on the genetic organization of *vmp* gene cassettes and the results of analyses of the segments from the expression cassette to the right end of the lp4 *vmp* expression plasmid (referred to as "expression locus segment" in this article) in reisolates obtained from infected mice. In STRF *Borrelia*, the expression *vmp* cassette on a linear plasmid is replaced by one of archival silent *vmp* cassettes distributed on other linear plasmids via gene conversion, in which sequences called UHS and DHS are thought to be involved [13]. In contrast, the expression *vmp* cassette on the lp4 plasmid is altered by the conversion of the expression locus segment, in which the expression locus segment is replaced by a duplicated segment of other plasmids carrying the cluster of archival silent cassettes (lp5, lp6, and lp10.1) (Fig 7). The replacing segment starts from one of the silent cassette and ends at the right end of the plasmid; thus

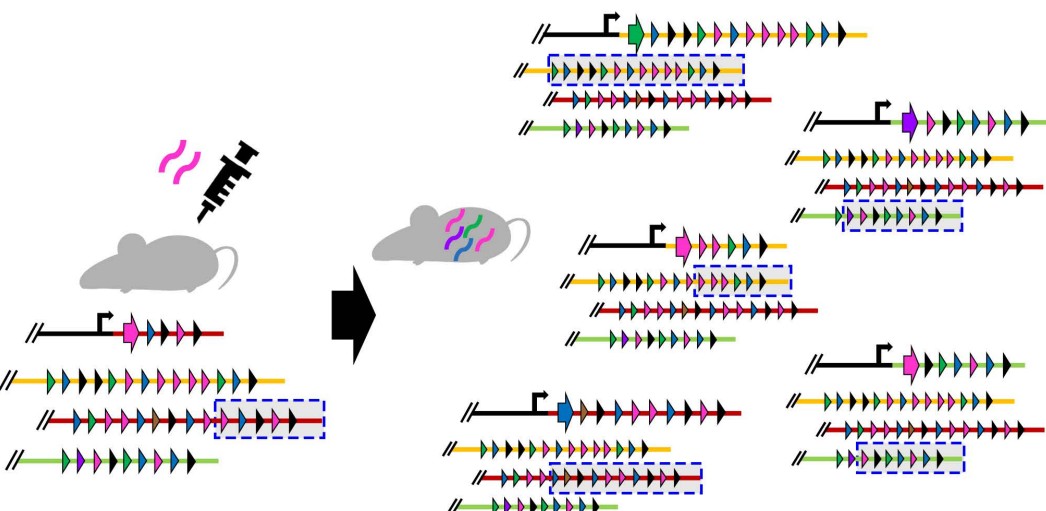

**Fig 7. The mechanism to change the expression *vmp* cassette in *B. miyamotoi*.** We demonstrated the dramatic changes in expression *vmp* cassette in *B. miyamotoi*. The clipart of mice in this figure was downloaded from open source resources (URL: https://www.ac-illust.com/).

the first silent cassette on the replacing segment is expressed after conversion. As the lengths of the replacing segment are valuable (ranged from 6.9 kb to 15.3 kb in the reisolates analyzed in this study; Fig 4), leading to the change in the length of lp4 (Fig 5). Although the molecular mechanism underlying the conversion of the expression locus segment is not known, it appears that the clustering of *vmp* cassettes and their tail-to-head cassette organization in each cluster allow the effective switching of the expression *vmp* cassette by the conversion of long plasmid segment. Notable structural features of *vmp*-encoding plasmids and *vmp* cassettes on these plasmids are the sequence conservation of the right ends of the lp4, lp5, lp6, and lp10.1 plasmids (S4 Fig) and the presence of UHS-like sequences around the start codon of *vmp* genes (S9 Fig). To understand the molecular mechanism underlying the conversion of the expression locus segment, it will be necessary to analyze the roles of these elements as well as identify the enzyme(s) involving the conversion.

## Materials and methods

### Ethics statement

Infection experiments were conducted in accordance with the Declaration of Helsinki and the Guidelines on Animal Experimentation established by the Institutional Animal Care and Use Committee of Yamaguchi University (Permission numbers: 242, 314, 386, 450, and 495).

### *Borrelia* strains and culture media

*B. miyamotoi* strains MYK1 (isolated from *I. persulcatus*), M1-2Br (re-isolated from the brain of B6 mice experimentally injected with strain MYK1), MYK2 (isolated from *I. persulcatus*), and MYK4 (isolated from *I. persulcatus*) were used in this study [42]. These *Borrelia* strains were grown at 30°C in modified Barbour–Stoenner–Kelly medium (BSK-M: using minimal essential medium alpha [Bio West, Germany]) supplemented with 10% rabbit serum under microaerophilic to anaerobic conditions, and the growth of spirochetes was examined using dark-field microscopy (at a magnification of 200×) [42]. Each strain was cloned by limiting dilution cultivation in 96 deep well plates containing BSK-M medium (1.8 ml) [3].

### Whole-genome sequencing

Genomic DNA of *B. miyamotoi* MYK1 clone G3 and M1-2Br clone H4 was purified using a Genomic-tip 100/G (QIAGEN, Tokyo, Japan) and used for preparing the libraries for sequencing using the PacBio RS II system (Pacific Biosciences, CA, USA) and Illumina reads (Illumina Inc., CA, USA). The PacBio reads were assembled using HGAP3, followed by error correction using Illumina reads as described previously [3]. The assembled sequences have been deposited in the DDBJ/EMBL/GenBank database (accession numbers are provided in S2 Table).

For the genome sequencing of MYK2 A9 and MYK4 C8, genomic DNA was purified using a Genomic-tip 100/G. Short-read libraries were prepared using the NEBNext Ultra II FS DNA Library Prep Kit for Illumina (New England Biolabs, MA, USA) and sequenced on the MiSeq platform to generate 300 bp paired-end reads (S5 Table). The short reads were trimmed using platanus_trim with the default parameters (http://platanus.bio.titech.ac.jp/pltanus_trim). Long-read libraries were prepared using the Rapid Barcoding Sequencing Kit (SQK-RBK004, Oxford Nanopore Technologies (ONT), Oxford, UK), sequenced on the MinION platform using an R9.4.1 flow cell (ONT), and base-called using guppy v5.0.16 (ONT). The long reads were trimmed using Nanofilt (quality score: > 10, read length: ≥ 2000 bp, initial 100 bp of reads trimmed, [43]). The trimmed long and short reads were assembled using micropipe [44] and the obtained DNA sequences were deposited in the DDBJ/EMBL/GenBank database (AP040247, AP040248, Bioproject ID; PRJDB20525).

### Genome finishing

Genome finishing of *B. miyamotoi* MYK1 clone G3 and M1-2Br clone H4 was performed by PCR and capillary sequencing of the PCR products and Illumina read mapping as previously described [3]. Capillary sequencing of PCR products was

performed to determine gap sequences likely due to tandem repeats and confirm single nucleotide polymorphisms. Capillary sequencing was performed using an ABI3130XL sequencer (Applied Biosystems, CA, USA) in Yamaguchi University or outsourced to Eurofins Genomics Inc. (Tokyo, Japan). Plasmid sequences were compared using GenomeMatcher ver. 3.10. [45].

## Mouse inoculation experiment

C57BL/6J (B6), BALB/cJ, BALB/c-nude (nude), and CB17SCID (SCID) mice were purchased from Jackson Laboratory (Kanagawa, Japan). Mice were inoculated via intraperitoneal injection with $2 \times 10^5$ bacteria in the BSK-M medium. For analysis of bacteremia using qPCR, 10 μL of whole blood was collected from the tail vein 5, 10, 15, 20 dpi for B6, BALB/cJ, and nude and 5, 10, 20 dpi for SCID, and DNAs were purified as described below. At the end of experiment, the mice were euthanized under anesthesia, and whole blood was collected via cardiac puncture. For expressing *vmp* analysis, 20–100 μL of whole blood or liver samples were inoculated to BSK-M medium. Five days post-inoculation, the supernatant of the culture medium was inoculated into new BSK-M medium and incubated under microaerophilic to anaerobic conditions.

## DNA purification of re-isolates and mouse liver tissue and whole blood

DNA of bacteria cultured in BSK-M medium were purified using a Wizard Genomic DNA purification kit (Promega Corporation, Madison, WI, USA). Ten microliters or 100 μL of whole blood collected from the tail vein or cardiac puncture, respectively, and a piece of liver tissue were used for DNA extraction using DNeasy Blood and Tissue Kit (QIAGEN).

## qPCR

Whole blood was assayed using qPCR with probes and primers specific for the borrelial *16S rRNA* gene. A minor groove binder probe with a VIC label (VIC probe) was designed for RF *Borrelia*, including *B. miyamotoi*, as described previously [42,46]. Two microliters of extracted DNA was subjected to qPCR using the Premix Ex *Taq* kit (Takara Bio Inc., Shiga, Japan). The average concentration of DNA extracted from whole blood was 0.86 μg/μl (measured by Qubit HR kit [Thermo Fisher Scientific, Waltham, MA, USA]). In this study, equal volumes of 2 μl were used for qPCR, and calculations were based on the original volume used for DNA extraction. qPCR was performed using a LightCycler 480 System II (Roche Diagnostics, Basel, Switzerland). The PCR cycles were set at 45, and the reaction was performed in 20 μL.

## Analysis of expression *vmp* cassettes in isolates

The parental and re-isolated strains from the mouse experiments were cloned by limiting dilution, and the clones obtained were analyzed for their expression *vmp* cassettes. DNA fragments containing the expression cassettes of each clone was amplified using the forward primer Vmp-UHR-F and reverse Vmp-DHR-R1 or Vmp-DHR-R2 primers using Tks Gflex DNA polymerase (Takara Bio Inc.) as follows: 95°C for 1 min, followed by 35 cycles of 95°C for 15 s, 55°C for 30 s, and 72°C for 1 min 30 s, and a final 5-min extension step at 72°C. The primer Vmp-UHR-F was designed upstream of the expression locus on the linear plasmid lp4 of the M1-2Br clone H4 (Figs 1 and S4). Reverse primers were designed for the intergenic conserved sequence between silent cassettes located on linear plasmids lp5, lp6, and lp10.1. In addition, a sequencing primer, Vmp-UP-60F, was used for sequencing expression cassettes. For expressing *vmp* analysis, total RNA was purified using the RNeasy Mini Kit (QIAGEN). After DNase I (Thermo Fisher Scientific) treatment, the mRNA of the expression cassette was amplified using the OneStep RT-PCR kit (QIAGEN). The following primer pairs were used to amplify *vlp*-D: vlp-66F-RNA and vlp-968R-RNA. The primers were designed to match the constant region of the *vlp*-D silent cassettes. The PCR products were sequenced at Eurofins Genomics, Inc. The primer sequence used in this study were listed in S6 Table.

## Detection of expression *vmp* cassettes in the DNA of mouse whole blood and liver tissue

DNA purified from the liver or whole blood of mice was used. The upstream forward primer Vmp-UHR-F and downstream reverse primers Vmp-DHR-R1 and Vmp-DHR-R2 were used to amplify the region containing the expression *vmp* cassette. PCR amplification was performed using Tks Gflex DNA polymerase under the PCR conditions described above. PCR products were sequenced after TA cloning. For the A-tailing of PCR products, PCR amplicons were used for the templates of nested PCR with Ex Taq DNA polymerase (Takara Bio Inc.). Nested PCR was performed under the same conditions as the Gflex PCR but with 15 cycles. PCR products of Ex Taq were ligated into the pGEM-T easy vector (Promega) and transformed into *Escherichia coli* strain DH5a (TOYOBO, Osaka, Japan or NIPPON GENE, Tokyo, Japan). The inserted fragments were amplified with Ex Taq DNA Polymerase using Sp6 and T7 primers and sequenced after purification using the AxyPrep MAG PCR Clean-Up kit (CORNING, NY, USA). The PCR products were sequenced at Eurofins Genomics, Inc.

## Length estimation of expression locus segments using long PCR

To estimate the lengths of expression locus segments, long PCR was performed using Tks Gflex DNA polymerase with forward primer, Vmp-UHR-F, and reverse primer, Vmp-LR, as follows: 95°C for 1 min, followed by 40 cycles of 95°C for 15 s, 60°C for 30 s, and 72°C for 5 min, and a final 5-min extension step at 72°C. Reverse primers were designed to match the constant region of the terminal end of lp4, lp5, lp6, and lp10.1. The electrophoresis was performed using 0.7% of SeaKem Gold agarose gels (Lonza, Basel, Switzerland) and DNA bands were visualized by Printgraph (ATTO, Tokyo, Japan) after incubation in an ethidium bromide solution (Nacalai Tesque, Inc., Kyoto, Japan).

## PFGE

PFGE was performed as previously described [2]. Briefly, bacteria were grown in BSK-M medium until the mid-log phase. Each strain was suspended in distilled water to a turbidity of 3.0 x 10^8 cells/300 µL. An equal volume of 1% SeaKem Gold agarose was mixed, immediately dispensed into plug molds (Bio-Rad, CA, USA), and treated with proteinase K (Roche Diagnostics). After treatment with 5 mM pefabloc SC (Roche Diagnostics) at 50°C, the plugs were stored in TE buffer at 4°C until use. The plugs were loaded onto 1.0% SeaKem Gold agarose gels. PFGE was performed with the CHEF-DRII System (Bio-Rad) using the following run parameters: a switch time from 0.1 to 12 s and a run time of 15 h at 6 V/cm. The gel was incubated in an ethidium bromide solution for 30 min for band visualization.

## Southern blot analysis

For Southern blot analysis of lp4 after PFGE, DNA was transferred to Hybond-N+ (Cytiva, Tokyo, Japan) membranes. After baking using UV, membranes were hybridized with the probe, which was labeled using Alkphos ECL Direct Labelling and Detection System with CDP-*Star* (GE Healthcare Japan) at 55°C overnight. The probe was targeted the upstream region of the *vmp* promoter and prepared by PCR amplification using Gflex DNA polymerase with the Probe_F and Probe_R primers. For Southern blot analysis of fragments containing the expression locus segment, the genomic DNA of each strain was digested by *Xba* I, *Eco*R I, *Spe* I, or *Bam*H I (Takara Bio Inc.) and separated by 1% STAR Agarose (RIKAKEN, Nagoya, Japan), and the DNA was transferred to the Hybond-N+ membranes. After baking using UV, membranes were hybridized with the probe, which was labeled using ECL Direct Nucleic Acid Labelling System (Cytiva) at 42°C overnight. The probe was targeted the upstream region of the *vmp* promoter and prepared by PCR amplification using PrimeSTAR GXL DNA Polymerase (Takara Bio Inc.) with the following primers: BmHJc13_Probe_F and BmHJc13_Probe_R. Clarity Western ECL Substrate (Bio-Rad) was used to detect labeled DNA, which was visualized using ImageQuant 800 (Cytiva).

## Mapping of the Illumina reads of the re-isolates and their parental strains to the M1-2Br H4 genome

For Illumina sequencing of re-isolate strains, genomic DNA was purified using Genomic-tip 100/G. Short-read libraries were prepared using the NEBNext Ultra II FS DNA Library Prep Kit for Illumina and sequenced on the MiSeq platform to generate 300 bp paired-end reads (S5 Table). Trimming was performed as described above. After masking the expression locus segment of M1-2Br H4 genome sequence (lp4, AP024359, 1–11,967 bp), Illumina reads were mapped to the M1-2Br H4 sequence using bwa-mem ver. 0.7.17 [47]. The mapping depth per base was calculated using SAMtools ver. 1.19 [48] and normalized using the mean depth of the specified region of each plasmid (AP024357: 29,852–31,927 bp, AP024361: 4240–5240 bp, AP024362: 6100–7100 bp, and AP024366: 2040–2689 bp). The mapping depths in 100-bp windows sliding every 10 bp were depicted in R ver. 4.2.3 [49] using the ggplot2 package ver. 3.5.0 [50]. The Illumina read were deposited in the DDBJ/EMBL/GenBank database (Bioproject ID; PRJDB20525).

## Western blot analysis

Whole-cell lysate of *B. miyamotoi* strain MYK2 clone A9 or MYK4 clone C8 were separated by 10–20% gradient gel (SuperSep Ace, Fujifilm Wako pure chemicals corporation, Osaka, Japan) and transferred onto a polyvinylidene difluoride membrane (Sequi-Blot PVDF membrane, Bio-Rad). The membrane was blocked in 5% Amersham ECL Prime Blocking Reagent (Cytiva) with PBS containing 0.1% Tween20. After blocking, membranes were incubated with mice sera (1:200 dilution) overnight at room temperature, followed by incubation with HRP conjugated goat anti-mouse IgG antibody (Thermo Fisher Scientific, 1:2,000). The reaction was detected using Clarity western ECL substrate (Bio-Rad) and visualized by the Amersham ImageQuant 800 (Cytiva).

## Supporting information

**S1 Fig. The comparison of plasmids of *B. miyamotoi* MYK1 G3 and M1-2Br H4.** The comparison of plasmid size and gene synteny of *B. miyamotoi* MYK1 G3 and M1-2Br H4 were shown using GenomeMactcher3.10. The high similarity value is shown in red and only results with homology >98% are shown in the figure. Upper and Lower indicate the MYK1 G3 and M1-2Br H4, respectively. A 4,593-bp relative deletion was found in lp4. In the MYK1 G3 strain, from Jc14 locus to upstream of Jc7 locus in the M1-2Br H4 strain was missing. The sequence of Jc7 and Jc14 was conserved at the first 180 bp of 5′ region, while the 3′ region was highly variable.
(TIF)

**S2 Fig. The phylogenetic analysis of *vlp* cassettes.** The unrooted (A) and rooted (B) Maximum Likelihood tree based on the Kimura 2-parameter model was constructed using sequences of *vlp* of soft-tick borne RF borreliae; *B. duttonii* Ly, *B. turicatae* 91E135 and *B. hermsii* DAH, indicated by purple, red and pink, respectively, and hard-tick borne RF borreliae; *B. miyamotoi* Yekat-1, LB-2001, FR64b and M1-2Br H4 indicated by light green, dark green, light blue and blue, respectively. The reference sequences were downloaded from GenBank database.
(TIF)

**S3 Fig. The southern blot analysis of lp4.** The southern blot analysis using a promoter sequence-specific probe for *B. miyamotoi* MYK1 G3 and M1-2Br H4 were shown. The PFGE gel was used for southern blot analysis of lp4. The promoter region of *vmp* locate at lp4 was labeled. This region was encoded on a single plasmid and the size difference between two strains was consistent with the results of the genome analysis. Molecular size markers that are denoted on panel in kilobase pairs.
(TIF)

**S4 Fig. The comparison of *vmp* cassette-bearing linear plasmids of *B. miyamotoi* M1-2Br H4.** The 5 plasmids, lp4, lp5, lp6, lp10.1, and lp11, which *vmp* cassette-bearing plasmids were compared using GenomeMactcher3.10. The high

similarity region was detected in the right end sequences of lp4, lp5, lp6, and lp10.1. The high similarity value is shown in red and only results with homology >90% are shown in the figure. Red and yellow arrowhead indicates the primer sites on lp4 of Vmp-UHR-F and Vmp-LR, respectively.
(TIF)

**S5 Fig. The genome comparison of *B. miyamotoi* M1-2Br H4, MYK2 A9 and MYK4 C8.** The chrDNA and *vmp* cassette-bearing plasmids of three strains were compared using GenomeMactcher3.10. The high similarity value is shown in red and only results with homology >95% are shown in the figure. Although the plasmid sequence of MYK2 A9 and MYK4 C8 were not verified in detail, the contigs carrying the silent cassettes were selected for comparison. It was suggested the gene repertoire of silent cassettes was conserved in three strains.
(TIF)

**S6 Fig. The schema of mice experiments.** The schema of mice experiments was shown. The clipart of mice in this figure was downloaded from open source resources (URL: https://www.ac-illust.com/).
(TIF)

**S7 Fig. The frequencies of expression cassettes detected in mice samples which injected MYK2 A9 strain.** The frequencies of expression cassettes were analyzed by TA cloning. The parental strain expressed Jk16 and the change in expression *vmp* genes was observed from 5 dpi in immunocompetent mice. The circled number indicate individual mouse and the number in the pie charts is the number of clones detected from a mouse. ND: Not detected.
(TIF)

**S8 Fig. The frequencies of expression cassettes detected in mice samples which injected MYK4 C8 strain.** The frequencies of expression cassettes were analyzed by TA cloning. The parental strain expressed Je16 and the change in expression *vmp* genes was observed from 5 dpi in immunocompetent mice. The circled number indicate individual mouse and the number in the pie charts is the number of clones detected from a mouse. ND: Not detected.
(TIF)

**S9 Fig. Alignment of 5′ prime of *vmp* expression locus with silent cassettes of *B. miyamotoi* M1-2Br.** The UHS of the *vmp* expression locus and each silent cassette on M1-2Br H4 plasmid sequencing were compared by *in silico* analysis. RBS; ribosome binding locus. Start; start codon. Conserved nucleotides are shown on a black background.
(TIF)

**S10 Fig. The southern blot analysis of *B. miyamotoi* parental and reisolates strain.** The southern blot analysis of *B. miyamotoi* parental strain and the reisolates was shown. The 4 restriction enzymes were used and the promoter region of *vmp* gene was labeled. The ① and ② indicate the re-isolated strains at 5 dpi injected MYK2 A9; B6M-1w A3 and B6M-1w C2, respectively. The ③ and ④ indicate the re-isolated strains at 5 dpi injected MYK4 C8; B6M-9L H6 and B6M-11w B1, respectively. Molecular size marker is denoted on the center of the panel in kilobase pairs.
(TIF)

**S11 Fig. Mapping of Illumina reads from reisolates and their parental strains to the lp1 of *B. miyamotoi* M1-2Br H4 genome.** The Illumina reads of reisolates and their parental strains were mapped to the genome of M1-2Br H4. The expression locus segment of the M1-2Br H4 genome sequence (AP024359, 1–11967 bp) was masked and used as the reference. Mapping depth per base was calculated and normalized using the mean depth of the specified region (29852–31927, indicating the gray color). Mapping depths in 100-bp window sliding every 10 bp were depicted in R ver. 4.2.3 with ggplot2 package ver. 3.5.0.
(TIF)

**S12 Fig. Western blot analysis of *B. miyamotoi* inoculated mice.** Seroconversion of mice inoculated with *B. miyamotoi* MYK2 clone A9 or MYK4 C8. Molecular mass is indicated on the left of molecular weight markers. Each lane showed the reaction of independent mouse serum collected at 5-, 10- and 28-days post-inoculation (dpi). The BSK medium inoculated mice serum collected at 10 dpi were used as control.
(TIF)

**S1 Table. Summary information of NGS of MYK1 clone G3 and M1-2Br clone H4.**
(XLSX)

**S2 Table. Plasmid profile and accession numbers of MYK1 clone G3 and M1-2Br clone H4.**
(XLSX)

**S3 Table. Re-isolated strains from C57BL/6J mice and number of clones analyzed.**
(XLSX)

**S4 Table. Strain list used in this study.**
(XLSX)

**S5 Table. Summary information of NGS of MYK2 A9 and MYK4 C8 strains and its re-isolates.**
(XLSX)

**S6 Table. Primers used in this study.**
(XLSX)

**S1 Data. Raw data of real-time PCR.**
(XLSX)

## Acknowledgments

The authors thank Ms. Ranna Nakao and Dr. Kozue Satoh (NIID) for providing information on *B. miyamotoi*. The authors are grateful to Ms. Junko Mizuno, Keiko Inagaki, and Ayako Hasegawa for their technical support. The authors would also like to thank Dr. Alice Lau Ching Ching (Yamaguchi University) for English editing of this manuscript.

## Author contributions

**Data curation:** Tomohi Takeuchi, Yasuhiro Gotoh, Hiroki Kawabata, Ai Takano.

**Investigation:** Tomohi Takeuchi, Yasuhiro Gotoh, Hiroki Kawabata, Ai Takano.

**Methodology:** Tomohi Takeuchi.

**Supervision:** Ai Takano.

**Visualization:** Yasuhiro Gotoh, Tetsuya Hayashi.

**Writing – original draft:** Tomohi Takeuchi.

**Writing – review & editing:** Tetsuya Hayashi, Hiroki Kawabata, Ai Takano.

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
