## [Decision Letter · Decision Letter 0]

13 Jul 2025

Antigenic variation is caused by long plasmid segment conversion in a hard tick-borne relapsing fever Borrelia miyamotoi

PLOS Pathogens

Dear Dr. Takano,

Thank you for submitting your manuscript to PLOS Pathogens. After careful consideration, we feel that it has merit but does not fully meet PLOS Pathogens's publication criteria as it currently stands. Therefore, we invite you to submit a revised version of the manuscript that addresses the points raised during the review process.

Please submit your revised manuscript within 60 days Sep 11 2025 11:59PM. If you will need more time than this to complete your revisions, please reply to this message or contact the journal office at plospathogens@plos.org. Please include the following items when submitting your revised manuscript:

We look forward to receiving your revised manuscript.

Kind regards,

Bersissa Kumsa, DVM, MSc, PhD

Academic Editor

PLOS Pathogens

David Skurnik

Section Editor

Editor-in-Chief

PLOS Pathogens

PLOS Pathogens

orcid.org/0000-0002-7699-2064

**Additional Editor Comments:**

Dear Authors,

The reviewers have completed their evaluation of your manuscript. I encourage you to revise and resubmit your work, ensuring that all reviewer comments are thoroughly addressed. Please incorporate the feedback carefully and provide a detailed, point-by-point response that clearly outlines every change made in response to the reviewers’ suggestions.

In addition, kindly correct all typographical and grammatical errors, and ensure that the manuscript is prepared in full compliance with the journal’s formatting and submission guidelines.

We look forward to receiving your revised submission

**Journal Requirements:**

1) We noticed that you used the phrase 'data not shown' in the manuscript. We do not allow these references, as the PLOS data access policy requires that all data be either published with the manuscript or made available in a publicly accessible database. Please amend the supplementary material to include the referenced data or remove the references.

2) Some material included in your submission may be copyrighted. According to PLOSu2019s copyright policy, authors who use figures or other material (e.g., graphics, clipart, maps) from another author or copyright holder must demonstrate or obtain permission to publish this material under the Creative Commons Attribution 4.0 International (CC BY 4.0) License used by PLOS journals. Please closely review the details of PLOSu2019s copyright requirements here: PLOS Licenses and Copyright. If you need to request permissions from a copyright holder, you may use PLOS's Copyright Content Permission form.

Potential Copyright Issues:

i) Figures 7, and S6. Please confirm whether you drew the images / clip-art within the figure panels by hand. If you did not draw the images, please provide (a) a link to the source of the images or icons and their license / terms of use; or (b) written permission from the copyright holder to publish the images or icons under our CC BY 4.0 license. Alternatively, you may replace the images with open source alternatives. See these open source resources you may use to replace images / clip-art:

3) We note that your Data Availability Statement is currently as follows: "The data that support the findings of this study are publicly available from DDBJ/EMBL/GenBank database with the Bioproject ID; PRJDB20525". Please confirm at this time whether or not your submission contains all raw data required to replicate the results of your study. Authors must share the “minimal data set” for their submission. PLOS defines the minimal data set to consist of the data required to replicate all study findings reported in the article, as well as related metadata and methods (https://journals.plos.org/plosone/s/data-availability#loc-minimal-data-set-definition).

4) Please amend your detailed Financial Disclosure statement. This is published with the article. It must therefore be completed in full sentences and contain the exact wording you wish to be published.

2) If any authors received a salary from any of your funders, please state which authors and which funders..

**Reviewers' Comments:**

Reviewer's Responses to Questions

**Part I - Summary**

Reviewer #1: This manuscript by Waugh et al. addresses the effects of Treponema pallidum exposure on the gene transcript levels and other cellular properties in immortalized human cortical endothelial cell cultures. While the manuscript contains a lot of data, the presentation of the data and its interpretation is overly complex, making it challenging for the reader to understand the results and verify the conclusions by examining the data themselves. Suggestions are made to try to improve the presentation and other aspects.

Reviewer #2: (No Response)

Reviewer #3: The authors provide a comprehensive analysis detailing the comparative repertoires and arrangements of vmp archival gene cassettes from the genome sequences of Japanese B. miyamotoi strains. The authors use this information to demonstrate antigenic switching from reisolates obtained from needle infected mice. Significantly, the authors provide new information showing that gene cassette switching involves replacing the expression locus and downstream cassettes with a long segment from another archival plasmid that is a different mechanism from soft tick relapsing fever borrelia variation. This report is an important advancement for analyzing the mechanisms of antigenic variation in this hard tick relapsing fever spirochete that was initiated in the seminal studies by the Hovius and Gilmore laboratories.

**Part II – Major Issues: Key Experiments Required for Acceptance**

Reviewer #1: 1. The Summary does not address the protein expression or cellular morphology results in Figs. 3 and 7.

2. The final paragraph of the Introduction contains information that should instead be described in the Results and Discussion.

3, The manuscript would benefit from a diagram figure showing the experimental procedure and time course. Important elements of the study (T. pallidum source used, number of T. pallidum, IEC control) are not described adequately. It is unclear why a medium control was not also included in the transcriptome analysis.

4. The Results contains an excessive amount of verbiage describing the transcript abundance trends, and refers to massive supplemental tables where it is nearly impossible for the reader to find the related data. It would be better to include additional figures in which data for some of the most important regulatory networks are extracted and clearly shown; the detailed text descriptions should be removed or reduced considerably. As an example, much emphasis is placed on increased Snai1 transcript levels; however, that information is buried in Table S1 and not easily accessible. A figure showing its expression and that of genes in its regulon would be helpful.

5. There seems to be a disconnect between the transcript results, in which significant changes are generally seen at 12 and 24 hours, and the cellular changes shown in Fig. 7, which are maximal at 1 hour and dissipate by 2 hours. This result seems to indicate that signal transduction and cellular changes precede the transcript level changes. This point needs to be better described in the Results and incorporated into the Discussion.

6. The term ‘expression’ is frequently used to describe transcript levels, which of course is not equivalent to either transcription (or degradation) rates or protein expression levels. This aspect is further amplified by a tendency to overinterpret the results. The manuscript should be revised fairly extensively to correct these issues.

7. l. 614. It is stated here that either T. pallidum preparations from infected rabbits or in vitro cultured T. pallidum were utilized to inoculate the hCMEC/d3 cell cultures. This is an important difference, in that the rabbit tissue extract would be expected to contain inflammatory products produced in response to the T. pallidum infection. Also, the RNA-seq results presented appear to be from a single experiment with 5 replicates per time point and VTP or IEC condition. Therefore, it is important to clearly indicate which preparation was used in each experiment.

8. l. 632. hCMEC/d3 is an immortalized cell line, not primary or secondary cultures of human cerebral endothelial cells. This aspect should be stated clearly here, in the abstract, and in the results section. Also, the possibility that the observed transcriptional responses of these immortalized cells may differ from those of cerebral endothelial cells that have not been immortalized should be addressed in the Discussion.

9. L. 640-644. It is first stated that the hCMEC/d3 cells were lysed by the RNAprotect treatment, and later indicated that the suspensions were centrifuged to pellet the cells. Please clarify. Also, perhaps it should be said that motility was used as a measure of viability of T. pallidum (rather than “observed for motility and viability”); it is unclear how viability would be examined by means other than motility.

10. The T. pallidum viability assessment results should be reported in the Results section.

11. The titles and descriptions for the supplementary tables should be included within each file, along with a key of abbreviations.

Reviewer #2: (No Response)

Reviewer #3: I have no concerns or issues regarding the data or the conclusions or interpretations thereof. The amount and presentation of the data is impressive, although some areas could be slightly modified to guide the readership better (see suggestions below). Overall, an important body of work that should be of interest to the research community.

**Part III – Minor Issues: Editorial and Data Presentation Modifications**

Reviewer #1: See above

Reviewer #2: (No Response)

Reviewer #3: Minor comments for consideration:

1. Paragraph beginning with line 127; would be helpful to include the GenBank accession number here for the genome sequence of the strains. The accession numbers given in the supplementary tables were difficult to find on the NCBI search (at least for me).

2. Line 247; this was an area of confusion regarding which supplementary Tables to check. Suggest referencing Table S3 on line 248 instead of having both S2 and S3 on line 252. Also suggest a separate column on Table S3 stating the original expressed vmp gene.

3. Figure 5; suggest arrows to indicate the bands with the expression plasmid.

4. Lines 400-401; add the reference Hoornstra et al., iScience 2024. Genome sequences of 21 isolates were documented.

5. Line 464; Is there a reference that defines the ingredients of BSK-M? I’m not familiar with this media or how if differs from BSK-II, BSK-H, or BSK-R. If not, consider adding the specifics here.

6. Line 522; “Two microliters of extracted DNA…”, replace microliters with micrograms to show quantity used.

PLOS authors have the option to publish the peer review history of their article (what does this mean?). If published, this will include your full peer review and any attached files.

Reviewer #1: No

Reviewer #2: No

Reviewer #3: No

**Figure resubmission:**

**Reproducibility:**



---

## [Editor Report · Decision Letter 1]

4 Sep 2025

Dear Prof Takano,

We are pleased to inform you that your manuscript 'Antigenic variation is caused by long plasmid segment conversion in a hard tick-borne relapsing fever Borrelia miyamotoi' has been provisionally accepted for publication in PLOS Pathogens.

Best regards,

David Skurnik, M.D., Ph.D.

Section Editor

PLOS Pathogens

Sumita Bhaduri-McIntosh

Editor-in-Chief

PLOS Pathogens

orcid.org/0000-0003-2946-9497

Michael Malim

Editor-in-Chief

PLOS Pathogens

orcid.org/0000-0002-7699-2064
---

## [Editor Report · Acceptance letter]

Dear Prof Takano,

We are delighted to inform you that your manuscript, "Antigenic variation is caused by long plasmid segment conversion in a hard tick-borne relapsing fever Borrelia miyamotoi," has been formally accepted for publication in PLOS Pathogens.

Best regards,

Sumita Bhaduri-McIntosh

Editor-in-Chief

PLOS Pathogens

orcid.org/0000-0003-2946-9497

Michael Malim

Editor-in-Chief

PLOS Pathogens

orcid.org/0000-0002-7699-2064